# Correction of Thoracic Hypokyphosis in Adolescent Scoliosis Using Patient-Specific Rod Templating

**DOI:** 10.3390/healthcare11070980

**Published:** 2023-03-29

**Authors:** Shivan Marya, Mahmoud Elmalky, Alex Schroeder, Anant Tambe

**Affiliations:** 1Royal Manchester Children’s Hospital, Oxford Road, Manchester M13 9WL, UK; 2Salford Royal Hospital, Stott Lane, Salford M6 8HD, UK; 3Faculty of Medicine, Menoufia University, Menoufia, Shebin Elkoum 6131567, Egypt; 4Digital Health Strategy, Stryker, 5404 Thornbrook Pkwy, Columbia, MO 65203, USA

**Keywords:** adolescent idiopathic scoliosis, thoracic hypokyphosis, patient-specific rod templating

## Abstract

The emphasis of surgical correction in adolescent idiopathic scoliosis (AIS) has been given to coronal plane correction of deformity without addressing the sagittal plane thoracic hypokyphosis. Thoracic hypokyphosis has been implicated in cervical malalignment, increased incidence of proximal and distal junctional kyphosis, spinopelvic incongruence, and increased incidence of low back pain. The surgeon, variability in surgical technique, and difference in rod contouring have been implicated as factors resulting in less-than-adequate restoration of thoracic kyphosis. We hypothesised that predictable correction of hypokyphosis could be achieved by using a reproducible surgical technique with patient-specific rod templating. We describe a technique of correction of AIS with dual differential rod contouring (DDC) using patient-specific rod templating to guide intraoperative rod contouring. The pre- and post-operative radiographs of 61 patients treated using this technique were reviewed to compare correction of hypokyphosis achieved with that predicted. Analysis revealed that we achieved a kyphosis within +/− 5.5 of the predicted value. The majority of patients had a post-operative kyphosis within the optimal range of 20–40 degrees. We concluded that patient-specific rod templating in DDC helps surgeons to consistently achieve sagittal correction in AIS close to a predicted value while achieving a very good coronal plane correction.

## 1. Introduction

Adolescent idiopathic scoliosis (AIS) is characterised by deformity of the growing spine that occurs in children between the ages of 10–16 years [1]. As the spine grows, it undergoes lateral deviation in the coronal plane, hypokyphosis in the sagittal plane, and axial rotation [2]. Traditionally, the aim of surgery in AIS was to achieve improvement of the lateral deviation deformity, with the surgical manoeuvres often resulting in a compromise of sagittal correction in favour of coronal correction [3]. More recently, however, the emphasis has shifted to achieve maximal correction in the sagittal, coronal, and axial planes while attempting to retain as much flexibility in the spine as possible [4].

Persistent thoracic hypokyphosis has been implicated in a loss of lumbar lordosis [5,6] with a resultant higher incidence of proximal (PJK) and distal junctional kyphosis (DJK) [3,7]. The literature also shows an increased incidence of lower back pain due to spinopelvic incongruence, which causes extensor muscle fatigue while maintaining upright posture [5,8], decreased pulmonary function with a thoracic kyphosis of less than 20 degrees [9], and cervical spinal kyphosis with a thoracic kyphosis of less than 26 degrees [10]. All these findings from studies suggest that restoration of thoracic kyphosis is an important goal in surgery for AIS.

Various manoeuvres for scoliosis deformity correction have been described as they evolved along with our understanding of the deformity itself [11]. The literature, however, suggests that there is considerably variability in the techniques employed and results achieved, and this variability is largely surgeon-dependent [12]. This has been attributed partly to free-hand rod contouring, which contributes to the variability.

In this paper, we describe a technique of pre-operative planning to make patient-specific templates for rod contouring that can be used to guide rod bending during surgery and achieve a balanced correction in both the sagittal and coronal planes. These templates serve as a guide for the surgeon to bend the rods intraoperatively in order to achieve optimal correction. This technique is routinely used for scoliosis deformity correction in patients with adolescent idiopathic scoliosis of Lenke types 1–4. We present the radiological outcomes to determine whether this templating helped achieve a predictable degree of correction in the sagittal plane, thereby reducing the variability associated with free-hand rod contouring.

## 2. Materials and Methods

All patients that underwent surgery at our centre performed by the senior author (ADT) from 2017–2021 with AIS Lenke type 1,2,3, and 4 curve patterns were included in this study. Patients with atypical curve patterns secondary to syndromic aetiology, congenital scoliosis, and neuromuscular scoliosis and those with Lenke type 5 or 6 curves were excluded. Appropriate consent was obtained from all parents and patients, and all procedures performed for the study complied with the ethical requirements set forth in the 1964 Declaration of Helsinki and its subsequent amendments.

Posteroanterior, lateral, and lateral bending full-length scoliosis radiographs with calibration markers were obtained within thirty days of planned surgical procedure and were formatted using stitched regional views, and a calibration marker was added to each image. The images were de-identified of all personal health information, given a unique identifier number, and loaded into a proprietary Surgimap (v2.3) software.

Measures were obtained using the Coronal and Sagittal Spinopelvic Wizard tools. Lenke curve types were determined based on standard parameters [13]. The upper and lower instrumented vertebra (UIV/LIV) were selected based on Cobb–Cobb, neutral vertebra, stable vertebra, and last-touched vertebra criteria. The UIV was also adjusted based on shoulder height on clinical examination. Age-adjusted sagittal normative values were generated by the software, and the rod contour and lengths were determined using a patient-specific rod tool. The tool employs a proprietary algorithm used to calculate the radius of curvature and length of the sagittal rod contours as well as the transition between thoracic kyphosis and lumbar lordosis. An overbend factor was added to the thoracic kyphosis (TK) [14] for the concave side rail to account for rod flattening to ensure the appropriate curvature and length for a predicted end point of the TK. The end point was confirmed using a modified secondary predictive equation to match the PI and LL within ±10 degrees [15]. A calibrated one-to-one ratio template was generated. The sterile template that was available on the OR table was used to contour the rods during surgery.

Patient were positioned prone on an OSI Allen Table. Intravenous anaesthesia was used to allow multimodal spinal monitoring to be used. A straight midline skin incision with bilateral paraspinal muscle dissection to expose up to the tips of the transverse processes (TP) was used. Interspinous and facet joint releases were carried out at all the levels except the UIV. Pontes osteotomies were performed across the apical 3–5 levels in patients with curves greater than 65° that did not reduce to less than 50° on pre-operative assessment, using bending films and traction films under anaesthesia if needed [16,17]. Further concave and convex side releases were performed, including base of transverse process osteotomies on the convex side.

Appropriate pedicle screws were inserted under image intensifier control at all the levels on the concave side. On the convex side, a proximal and distal foundation of two screws was established, and subsequently, screws were inserted into strategic vertebra.

The dual differential rod contouring (DDC) technique with two-rod simultaneous translation was used in all cases [11,18]. This DDC technique has always been used by the senior authors in the correction of AIS. The only difference was the additional use of patient-specific templates. The patient-specific rod template was printed on A3 paper and placed in sterile clear plastic wrapping on the instrument trolley. This was used to guide contouring of both the convex rod and concave rail. The under-contoured convex rod was inserted first and partially locked into the proximal 2 pedicle screws. A cantilever technique was used to sequentially reduce the rod from proximal to distal, allowing it to sit flush with the screw tulips without being reduced completely into them at all the other levels. This rod acts as the pivot for vertebral body rotation and translation while inserting the concave rail. A 5.5 mm over-contoured titanium transitional rail was used on the concave side, as shown in the template, for all patients. The rail was first captured and fixed in the proximal and distal screw heads. It was then reduced into the screw heads sequentially with gradual increments. During the reduction, the rail was maintained co-axial to the screw heads to facilitate rail capture. The rail was reduced gradually to sit flush to the screw heads. Once the rail was captured at all screw levels, final correction was carried out using a combination of vertebral de-rotation compression and distraction to achieve good coronal and sagittal plane correction.

Pre-operative and 6-month post-operative radiographs were used to measure the correction achieved in both the coronal and sagittal planes. Comparisons between the change in magnitude as well as the percentage change of the Cobb angle were recorded. Pre- and post-operative coronal alignment was assessed with C7 plumb line, clavicle angle, T1 tilt, central sacral vertical line, and C7 offset, sacral and pelvic obliquity. The sagittal alignment measures included pre- and post-operative pelvic parameters, thoracic kyphosis, lumbar lordosis, sagittal vertical axis (SVA), and T1 pelvic angle (TPA). The sagittal alignment target parameters were PI-LL = ±10°, SVA < ±2 cm, PT < 20°. TK was evaluated both for change from pre-operative measure as well as accuracy of predicted TK (TKp). We also analysed how many patients with a TK of less than 20 degrees pre-op improved to more than 20 degrees post-op. Statistical analysis for each of the reported measures and summary includes means and standard deviation. Paired *t*-tests were used to compare pre-operative and post-operative values. Findings were considered significant for *p* < 0.05. Analysis was completed using JMP^®^, Version 17.0.0. (SAS Institute Inc., Cary, NC, USA, 1989–2023).

## 3. Results

There were 61 patients (53 female and 8 male) with a mean age of 14.54 (range 12–21). The average no of levels fused was 12.10 (range 7–15). The mean follow-up was 33.07 months (11–68 months). The hospital stay average was of 4.97 days (3–9 days).

Patient demographics are summarised in Table 1.

### 3.1. Coronal Plane Correction

The mean percentage changes for coronal plane correction in the proximal thoracic, main thoracic, and thoracolumbar curve correction were 48.1% ± 22.5, 75.4% ± 11.3, and 67.4% ± 28.0, respectively. All three parameters had a statistically significant change post-operatively when compared using the paired *t*-test. Coronal plane correction has been summarised in Table 2.

In total, 92% of patients (56 of 61) had sufficient data for analysis of the clavicle angle. There was a definite elevation of the left shoulder in the patients post-operatively, as shown by a change in the clavicle angle by an average of 5.5°and the T1 tilt by an average of 7.9° (Table 3). The P value for both measurements was <0.001 when analysed using the paired *t*-test.

### 3.2. Sagittal Plane Correction Parameters

All four sagittal measures, PI-LL, PT, SVA, and TPA improved post-operatively (Table 4). The change was significant using the matched-pair-*t* test for all parameters other than SVA. Using the Surgimap algorithm, the predicted thoracic kyphosis was 33.1° ± 6.1. The difference between the achieved kyphosis and the predicted kyphosis (prediction delta) was 5.3° ± 4.4.

Further analysis of the change in thoracic kyphosis was achieved by categorising patients were into three groups based on the pre-operative TK value (Table 5) as described by Rampal et al. [19]. As per this classification, pre-operatively, 7 patients were hypokyphotic (TK < 20°), 33 normo-kyphotic (TK 20–40°), and 21 patients were hyperkyphotic (TK > 40°). This revealed that post-operatively, there was a significant change in TK in both the hypo- and hyperkyphotic patient groups as a result of which these patients achieved mean TK within the ‘normal’ parameters of 20–40°. The normo-kyphotic group patients had a marginal, non-significant increase in TK post-operatively.

Figure 1 shows a graphical representation of pre- versus post-operative kyphosis. The majority of the patients have post-operative TK within the 20–40° range.

Figure 2 shows improvement in kyphosis post-operatively in all the patients who had less than 20 degrees of kyphosis pre-operatively. The graph shows that kyphosis was restored in most of these patients to within a normo-kyphotic range of 20–40 degrees.

Figure 3 represents the normo-kyphotic group of patients whose mean TK increased marginally and remained within the normal range.

Figure 4 shows the hyperkyphotic group of patients whose TK decreased significantly post-operatively with a mean post-operative TK value within the normal range.

### 3.3. Comparison of TK Achieved with Templating

The mean TK predicted using templating was 33.46 and post-operative TK achieved was 33.07. The mean prediction delta was 5.5°. The paired *t*-test of predicted TK and post-operative TK had a p value of 0.4, indicating there was no significant difference between the two. ANOVA showed a significant correlation between the predicted TK endpoint and the actual post-operative TK (*p* < 0.0001).

There were no patients with permanent neurological deficits post-operatively. However, in two patients, there was a loss of motor signals during the correction. The signals returned to normal when the correction was undone. Final correction was achieved with under-contouring the rod as compared to the template. These patients had thoracic kyphosis of less than 20 degrees post-operatively. None of the patients had PJK or DJK and none have had any form of revision surgery.

Figure 5, Figure 6 and Figure 7 are pre- and post-operative radiographs and rod templates of representative cases from the hypo-, normo-, and hyperkyphotic group of patients, respectively, who underwent surgical correction using this technique.

## 4. Discussion

The restoration of thoracic kyphosis seems essential to ensure good long-term outcomes following AIS surgery [5,6,7,8,9,10,20,21]. The appropriate amount of thoracic kyphosis for individual patients can be predicted using the formula suggested by Cement et al. [22]. However, Pesenti et al. stated that the formula had moderate accuracy, especially in patients older than 35 years of age [23]. Though prediction of the exact desired kyphosis may not be possible, an optimal amount of kyphosis restoration might help in avoiding all problems associated with persistent thoracic hypokyphosis. Rothenfluh et al. [24], in their retrospective analysis of 86 patients with Lenke 1 and 2 curves, found that achieving 23 degrees of kyphosis or more helps decrease the risk of sagittal plane decompensation following selective thoracic and thoraco-lumbar fusions. To avoid post-operative cervical malalignment and lordosis, achieving a minimum kyphosis of 26 degrees has been suggested [10].

Various factors have been implicated in thoracic hypokyphosis and reciprocal loss of lumbar lordosis after AIS surgery. They include:The surgeon themselves: Variation in surgeons’ choice of techniques and differences between different surgeons’ techniques, including variation in rod contouring [12,25,26,27,28,29,30].Implant density and implant choice: Higher screw density on the concave side with thicker and cobalt chrome rods have been associated with restoration of kyphosis to greater than 20 degrees [31]. Reduction in the implant density, however, also means the load exerted on each anchor is increased. One simulation study showed that the pull-out force exerted on the screw in the apical vertebra increased 2.5-fold when screw density decreased from 2.0 to 1.0 [32].Surgical correction technique: Dual differential rod contouring has been shown to achieve the best vertebral rotation [11,18]. Thoracic kyphosis is best restored through the dual rod posteromedial correction technique [33,34]. Manoeuvres such as simple rod roll and direct vertebral de-rotation have been found to worsen hypokyphosis as they are inherently lordogenic in nature [35,36].Use of osteotomies [16,17]: Newton et al. concluded [3] that to maintain thoracic kyphosis and ensure maintenance of lumbar lordosis, posterior column lengthening using pontes osteotomies is recommended.

A variety of surgical techniques using free-hand surgeon-dependent rod bending have been described. Some of them are able to restore thoracic kyphosis well. A recent analysis [37] concludes that there is improvement in our ability to restore thoracic kyphosis by posterior-only approaches as more surgeons are using a combination of the techniques above. This, again, highlights that one of the main factors in restoring thoracic kyphosis is based on the surgeon’s choice of techniques [25,26]. Despite improving results, not all surgeons using a concave side correction technique are able to consistently restore thoracic kyphosis to >20 degrees [31,38,39]. Gehrchen et al., using a concave-sided correction technique, compared round rods to a rail. They reported a 62% correction in the coronal plane using a rail. Though they maintained kyphosis in all cases more than more than 20 degrees, there was a reduction in kyphosis, which was less when they used a rail. However, they did not use any templates. We have used the rail in our surgical technique. Tsirikos et al. [40], using a novel convex side-based pedicular screw technique, reported upper thoracic scoliosis correction by a mean 68.2%, main thoracic scoliosis correction by a mean 71%, and lumbar scoliosis was corrected by a mean 72.3%. No patients lost more than 3° of correction at follow-up. The thoracic kyphosis improved by 13.1° to a mean value of 45.1 degrees; the lumbar lordosis remained unchanged. Using this technique, there was a tendency for post-operative hyperkyphosis (TK > 40). Other similar convex side screw-based techniques have been unable to restore thoracic kyphosis and have reported a loss of scoliosis correction of 5% to 7% at follow-up [40,41].

Regardless of the technique used to correct scoliosis deformity, the ultimate shape of the spine is dependent on the shape of the rods. Rod contouring plays a crucial role in the correction achieved, and a recent study has demonstrated poor reliability and high variability of free-hand rod bending [12,29,30]. The authors [29] found that even in experienced surgeons, free-hand bending results in overbending rods by a mean value of 18.9°. However, using a rod template, this can be improved to an average under-bending of rods by 0.2°. In addition, Abelin et al. [20] also stated that rod contouring has to be very patient-specific and dependant on the curve pattern to achieve a good correction. This variability or unpredictability in rod bending can be addressed by either having patient-specific rods that are manufactured by the implant company or by creating templates that the surgeon uses as a guide for contouring. Rampal et al. [42] published their findings of using patient-specific rods, and found that they helped to consistently achieve a predicted value of correction. However, the authors did not delve into the cost of manufacturing the pre-bent rods. Unlike their study, where they used commercially made patient-specific rods, we used patient-specific templates printed on A3 paper, as described above. This was used to help contour the rod and, hence, did not add any extra costs to the procedure.

The average hospital stay for our patients was 5.38 days, which was comparable to the average stay in other studies. In our cohort, we used dual differential rod contouring (DRC) in all cases, with the concave rail being over-contoured by 20 degrees [11,14,43,44]. This technique represents an additional correction strategy involving over-contouring of the concave rail to pull the concave side of the curve posteriorly, and under-contouring of the convex rod to push the convex side of the curve anteriorly and achieve a simultaneous correction in three planes. Previous in vivo and biomechanical studies have demonstrated the efficacy of DRC in achieving 3D deformity correction and have demonstrated the association of increased over-correction of the concave rod with increased amounts of sagittal plane correction and the ability to achieve correction in both the instrumented thoracic spine and in the un-instrumented lumbar spine [14,18,43]. Kluck et al. commented that [44] differential rod contouring in DRC is, by nature, subjective. Concave and convex rod contours are determined intraoperatively by the treating surgeon based on estimates of the desired sagittal profile and severity of the axial plane rotation. The ideal rod contours remain unknown. This highlights the need for patient-specific pre-templated rod templates, which would avoid the guesswork. In addition to over-contouring of the concave rod, we used an extra-strong concave rail, which would deform less, more concave side anchors, simultaneous correction using two rods with resultant posteromedial translation, and pontes osteotomies if needed. All these techniques have been proven to be important factors in restoring or maintaining thoracic kyphosis. One of the risks of using pontes osteotomies and an over-contoured rail is the potential for neurological injury. In our series of patients, there were no neurological injuries; however, two patients had changes in intra-operative monitoring. Hence, for these patients, we under-contoured the rods and accepted less correction in the sagittal plane compared with that predicted pre-operatively.

Our analysis of results revealed the following salient features of using pre-operative templating with the above-mentioned correction technique:Restoring TK to a value of 20–40° regardless of pre-operative TK value.Post-operative TK achieved was within 5.5° of the value predicted using templating.Excellent coronal plane correction in addition to the correction of TK.

Using our technique, we were able to achieve a 75.4% curve correction of the main thoracic curve in the coronal plane. In our study, the TK increased in all patients who were hypokyphotic, and decreased in those who were hyperkyphotic and were within the range of 20–40 degrees in all but two patients. The template helped us not only achieve a predicted amount of kyphosis based on the patient’s pelvic parameters but also restore a majority of patients to the normo-kyphotic range.

There are certain draw backs to our study. We do not know yet if predictable restoration of thoracic kyphosis will translate into a long-term clinical improvement. Lenke et al. did not find a clinical difference in the short term [45]. We used the overbend factor based on work by Cidambi et al. [43], which was using a COCR rod. With more experience with a titanium rail, we will be able to build in a more specific overbend based on bending properties of this rod, which would improve the accuracy of the template and ensure a more predictable kyphosis restoration.

## 5. Conclusions

Much like other fields of surgery with templating and patient-specific planning, the long-term clinical implications are yet to be established and proven. However, as demonstrated in this paper, it is possible to achieve a predictable degree of correction in the majority of cases. This technique does not add any extra cost, is reproducible, and does not increase the hospital stay or rehabilitation time. The authors feel that patient-specific templates can help guide surgeons intra-operatively, rather than relying on free-hand ‘eyeballing’ of the curve to contour the rods. This will help restore thoracic kyphosis to the predicted values based on pelvic indices and also restore a majority to the normo-kyphotic range.

## Figures and Tables

**Figure 1 healthcare-11-00980-f001:**
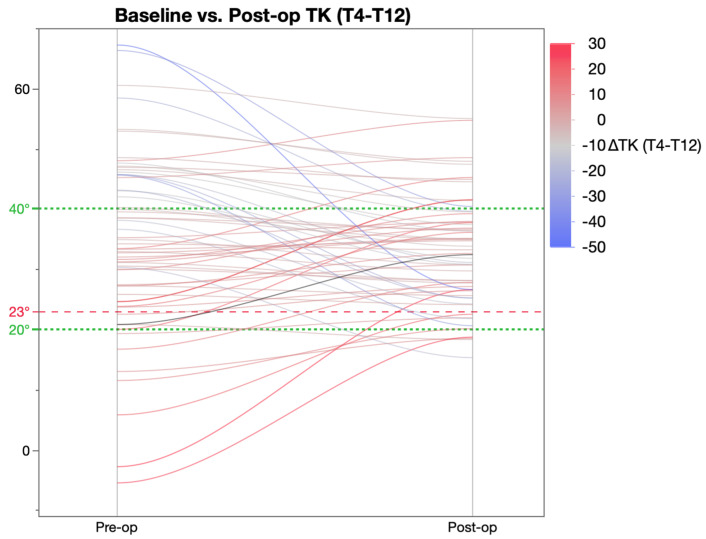
Pre and post-operative thoracic kyphosis measured on radiographs.

**Figure 2 healthcare-11-00980-f002:**
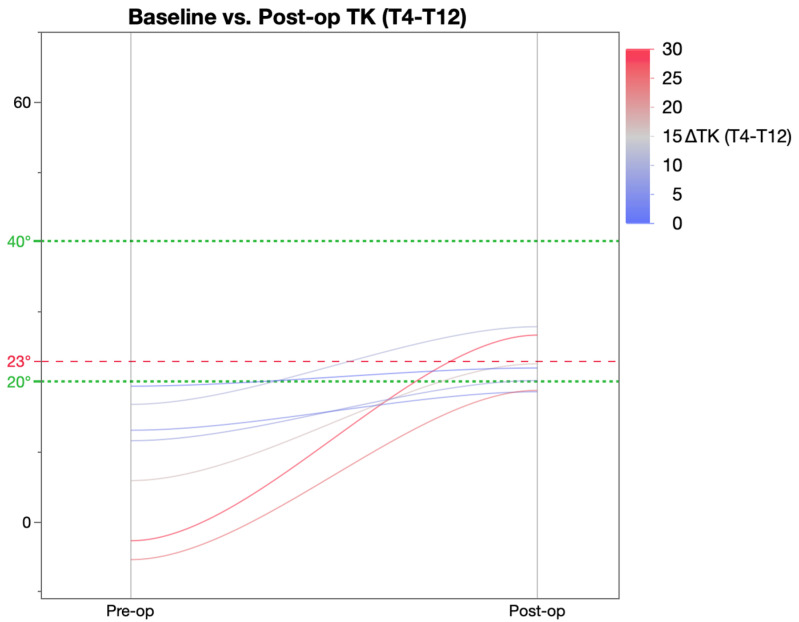
Pre and post-op kyphosis in patients with baseline TK less than 20 degrees.

**Figure 3 healthcare-11-00980-f003:**
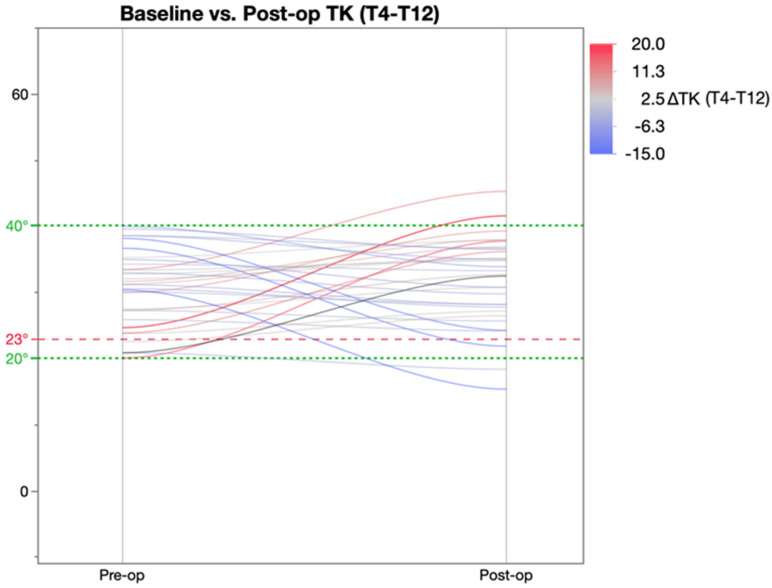
Pre and post-op kyphosis in patients with baseline TK of 20–40 degrees.

**Figure 4 healthcare-11-00980-f004:**
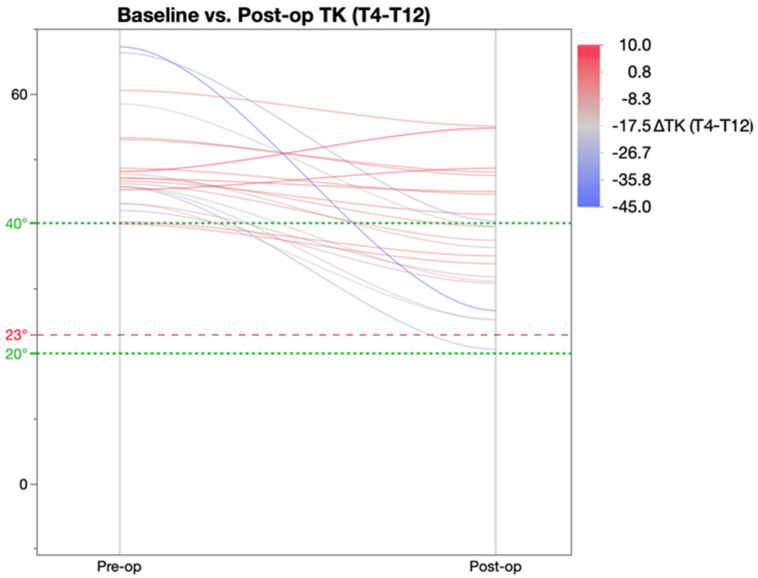
Pre and post-op kyphosis in patients with baseline TK more than 40 degrees.

**Figure 5 healthcare-11-00980-f005:**
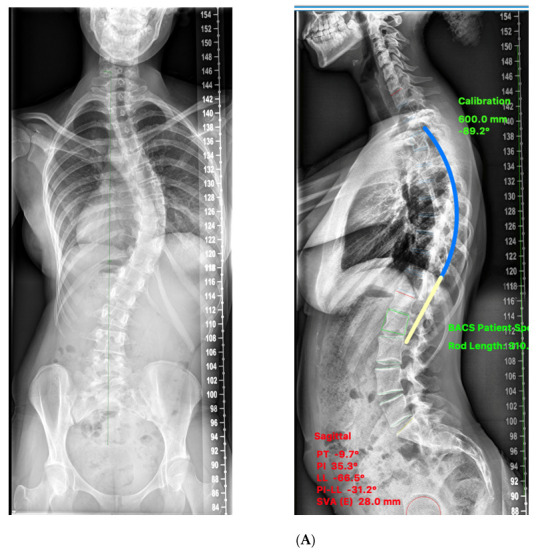
(**A**) Pre-op X-ray AIS F: 15 yrs. Pre-op TK 33, MT Cobb: 61.5. Predicted TK: 34. (**B**) Patient-specific rod template. (**C**) Post-op X-ray at 6 months. Post-op: TK: 38.6: MT: 1.5.

**Figure 6 healthcare-11-00980-f006:**
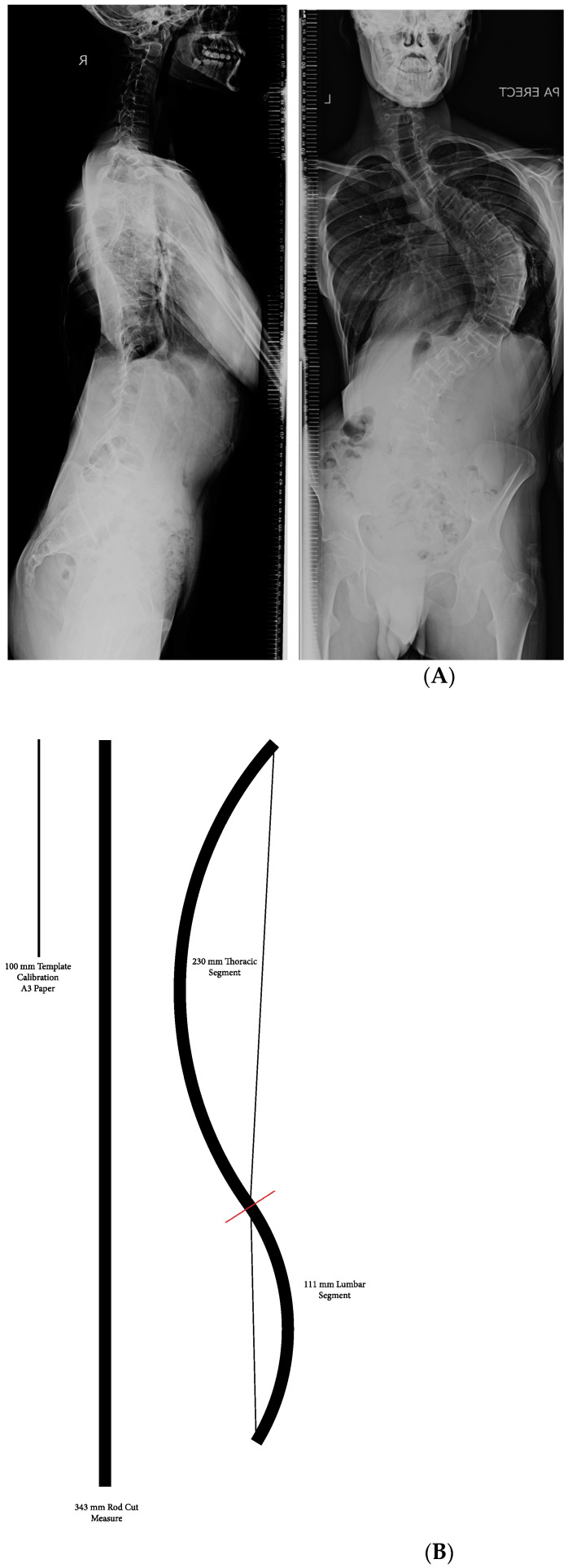
(**A**) Pre-op TK: −7.3 degrees (hypokyphosis), predicted TK: 20 degrees, pre-op MT Cobb: 98 degrees. (**B**) Templating using Surgimap and patient-specific template. (**C**) Post-op X-ray at 6 months. Achieved TK: 26 degrees, MT Cobb Post-op: 30 degrees.

**Figure 7 healthcare-11-00980-f007:**
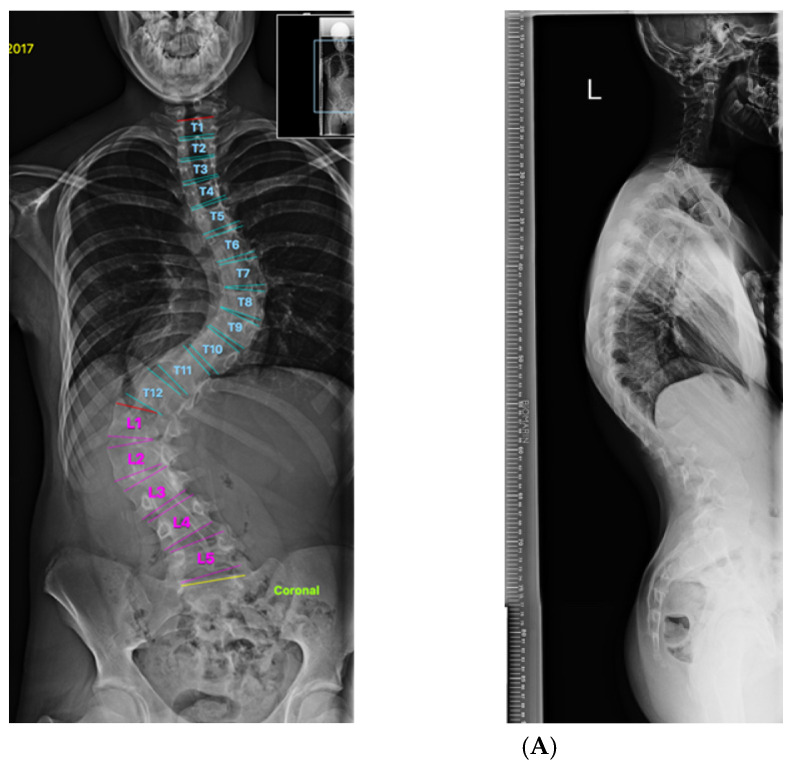
(**A**) Pre-Op MT C obb:65, TL-L Cobb: 78.5, TK: 54 (hyperkyphosis). Predicted TK 45. (**B**) Patient-specific template. (**C**) Post-Op Mt Cobb: 27, TL-l Cobb: 31 TK:45.

**Table 1 healthcare-11-00980-t001:** Patient demographics.

Total Patients	61 (53F)
Lenke 1	29
Lenke 2	13
Lenke 3	7
Lenke 4	12
Lumbar Modifier	
A	31
B	8
C	22
Thoracic Modifier	
−	5
N	43
+	13

**Table 2 healthcare-11-00980-t002:** Coronal plane curve correction.

Measurement	Pre-op Angle	Post-op Angle	Change in Angle	% Change	*p* Value
Proximal Thoracic	30.5° ± 10.2	15.6° ± 7.1	14.9° ± 7.9	48.1% ± 22.5	<0.001
Main Thoracic	68.5° ± 13.4	17.4° ± 9.0	51.2° ± 10.2	75.4% ± 11.3	<0.001
Thoraco–Lumbar	43.4° ± 14.2	15.5° ± (9.1)	28.6° ± 12.1	67.4% ± 28.0	<0.001

**Table 3 healthcare-11-00980-t003:** Shoulder alignment.

Clavicle Angle	
Pre-op	−2.5°± 3.2°
Post-op	3.0° ± 2.7°
*p* Value	<0.001
**T1 Tilt**	
Pre-op	−2.4(+/−8.81)
Post-op	5.42(+/−5.2)
*p* Value	<0.001

**Table 4 healthcare-11-00980-t004:** Sagittal plane correction parameters.

	Pre-op	Post-op	Mean Difference	*p* Value
Thoracic Kyphosis (all)	34.6° ± 14.6°	33.1° ± 9.0	–1.4 ± 12.7	
PI-LL	−10.7	−6.5	−4.3	0.0074
PT	7.8° ± 8.2°	10.8° ± 9.3°	2.6	0.0012
SVA	–13.5 ± 36.7	–6.9 ± 37.1	6.7	0.3087
TPA	3.0 ± 8.2	6.2 ± 9.6	2.8	0.0061

**Table 5 healthcare-11-00980-t005:** Subgroup analysis of TK pre- and post-operatively.

	N	Pre-op	Post-op	Mean Difference (∆TK)	*p* Value
Thoracic Kyphosis (all)	61	34.6° ± 14.6°	33.1° ± 9.0	–1.4°	
TK Hypokyhphosis group	7	8.4° ± 9.5°	22.4°± 3.7°	14.0°	0.0096
TK Normo-kyhphosis group	33	30.5° ± 5.7°	32.1° ± 6.8°	1.62°	0.2731
TK Hyperkyhphosis group	21	49.2° ± 7.7°	38.2° ± 9.5°	−11.0°	0.001

## Data Availability

The data presented in this study are available on request from the corresponding author. The data are not publicly available due to confidentiality.

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
