# Peer review of "Correction of Thoracic Hypokyphosis in Adolescent Scoliosis Using Patient-Specific Rod Templating"

_healthcare, 2023, doi:10.3390/healthcare11070980_

Round 1

Reviewer 1 Report

Dear Dr.,

Title: Correction of thoracic hypokyphosis in Adolescent Scoliosis using Patient Specific Rod templating

Manuscript ID: healthcare-2250704

Overall comments: Authors described in this manuscript: the predictable correction of hypokyphosis effect of reproducible surgical technique with specific rod templating in scoliosis patients. Further, the authors also described the possible sagittal and coronal plane correction in the scoliosis patient with a specific rod templating method. The overall manuscript is written well and it has novelty in this field of research. However, some of the typographical errors and sentences need to revise in this manuscript.

Specific comments:

1.      The title of this manuscript; the words ……thoracic hypokyphosis……….. can be change as ………Thoracic Hypokyphosis………...

2.      The abstract is well written.

3.      Line 122 to 126 statements of statistical methods applied is not clear. Need to rewrite it with more clear information.

4.      The introduction can be elaborate with applications of the Specific Rod templating technique. Further, can incorporate the background information of the inclusion and exclusion criteria of this study.

5.       Table 2. Value is bold; need to follow the uniform pattern as per instructions. The author needs to ensure that all tables are made correctly.

6.      Line 254 references need to place correct patterns.

7.      The discussion section has too many small paragraphs. Need to make logical arrangements.

8.      The author contribution section pattern needs to change.

9.      References are placed beyond the last 3 years; need to place the recent references corresponding to relevant text information.

*****

Author Response

We, the authors of this paper titled ‘Correction of Thoracic Hypokyphosis in Adolescent Scoliosis using Patient Specific Rod templating’ would like to thank the reviewer for their time in reading our paper and recommending changes.

  1. Thank you for the recommendation, We have changed the case of the letters in the title
  2. Thank you for your input
  3. We have changed this to provide a bit more information, it now reads:

‘Statistical analysis for each of the reported measures and summary includes means and standard deviation. Paired t-Tests were used to compare pre-operative and post-operative values. Findings were considered significant for p < 0.05. Analysis was completed using JMP¬Æ, Version 17.0.0. (SAS Institute Inc., Cary, NC, 1989-2023).’

  1. We have added the following:

‘These templates serve as a guide for the surgeon to bend the rods intraoperatively in order to achieve optimal correction. This technique is routinely used for scoliosis deformity correction in patients with adolescent idiopathic scoliosis of Lenke type 1-4.’

  1. Table formatting changed as suggested.
  2. We have changed the order of the references to numerical order
  3. Discussion section changed as per recommendations of reviewer 1 and 2
  4. Pattern of author contributions changed as per author guidelines
  5. References- a lot of the seminal scoliosis work is from prior to 3 years, however with respect to sagittal parameters and their newly described implication we have tried to quote the most recent literature in our paper.

Reviewer 2 Report

Reviewer’s comments:

 Marya et al. presents a patient-specific rod templating approach for thoracic hypokyphosis correction and the results they obtained in the patients that received this treatment.  Similar approaches have been reported and reviewed previously (PMID: 28612190, 31757655, and 33355708).  However, the findings in this current study can add on to this developing approach, and therefore, can be accepted for publication with minor revisions.  The detailed suggestions are listed below:

1.      In Line 40, the word “shows” appear twice, flanking the word also, please delete one at your preference.

2.      In Line 40, “low back pain”, I think the authors mean “lower back pain”.

3.      In Table 2, please comment on why statistical analysis are not performed.  If there is not a valid reason, please conduct the analysis and include the calculated p values.

4.      Similar to above, in Table 3, please comment on why statistical analysis are not performed.  If there is not a valid reason, please conduct the analysis and include the calculated p values.

5.      In Table 4, the last column in the first row states (P<= 0.05), however, the p value in the row of SVA is above 0.05 (0.3087).  Please revise the table so resolve the contradiction.  

6.      Similarly, in Table 5, the last column in the first row states (P<= 0.05), however, the p value in the second last row is above 0.05 (0.2731).  Please revise the table so resolve the contradiction.  

7.      In Line 195, the authors discussed Figure 5 and 7, while skipped Figure 6.  If the authors wish to discuss the data presented in Figure 7 before Figure 6, please rearrange the order, which is the norm in the published biomedical articles.

8.      The labeling in Fig. 5 is not legible.  Please enlarge the font. 

9.      Are the images and diagram on Page 10 part of Fig. 7?  Please label them properly. 

10.  The labeling in the diagram on Page 10 is not legible.  Please enlarge the font. 

Author Response

Thank you for your time to review our paper and the changes suggested. The reviewer is correct that there has been recent literature examining the roll of sagittal plane correction in scoliosis, however the papers they have provided the PMID for have looked at techniques and implants different to what we describe. While all aim to achieve sagittal correction, the unique aspect of our technique is that once the template is prepared using the software, it only requires printing on a paper that can be kept in a sterile wrapping thereby making it more widely applicable than ordering expensive implants from the manufacturers.

1,2- Thank you for recommending, we have corrected the errors

3-4- We have added the p values to both table 2 and 3 as suggested by the reviewer

5-6- Once again thank you for noticing these errors, we have corrected the title from ‘p<0.05’ to ‘p value’ in tables 4 and 5

7-10- Figures 5,6,7 have been formatted to address the issues highlighted

Reviewer 3 Report

Dear Aurors,

Congratulations on your well-prepared work and excellent clinical results.

As a reviewer, however, I have a duty to point out some shortcomings. In order to plan the patient's treatment, apart from the description of the method used and its effectiveness, some additional information is needed that is missing in the text.

1. Does the described method increase the operation time? This information is particularly important for the anesthetist.

2. How much does the cost of intervention increase when using the described solution? This is important for the patient and for the insurer.

3. Does the applied solution affect the recovery time? Pain after surgery (intensity and duration), rehabilitation process (limitations and duration), limitations in everyday functioning (durability of applied stabilization and healing time).

This is important information and supplementing it (albeit in a few sentences) will significantly improve the reception of the presented work and allow it to interest a wider audience.

Author Response

We, the authors of this paper titled ‘Correction of Thoracic Hypokyphosis in Adolescent Scoliosis using Patient Specific Rod templating’ would like to thank the reviewer for their time in reading our paper and raising pertinent questions. In response to the queries, we would like to say:

  1. The technique does not change the surgical time as the template is printed on an A3 paper and placed in sterile wrapping prior to starting the surgery. The template if at all hastens the rod bending step of surgery as it eliminates estimation of bend required for correction and makes it more accurate.
  2. Access to the software was available at no additional cost to our institution as the implant manufacturer provides this. From a resource point of view all that was needed was a print out of the template on an A3 paper wrapped in sterile clear plastic sheet so there was no substantial additional cost to using this technique.
  3. The senior author employs this technique to all Type 1-4 Adolescent idiopathic scoliosis correction surgeries. We have added data on length of stay to demonstrate that this is no different to not using rod templates. The surgery is in no way different to what was done before, except the pre operative planning and rod bending technique used.

We have incorporated the above into our manuscript too.